# Revisiting Hidden Representations in Transfer Learning for Medical Imaging

**Dovile Juodelyte**                                                                            *doju@itu.dk*
*IT University of Copenhagen, Denmark*

**Amelia Jiménez-Sánchez**                                                                      *amji@itu.dk*
*IT University of Copenhagen, Denmark*

**Veronika Cheplygina**                                                                         *vech@itu.dk*
*IT University of Copenhagen, Denmark*

**Reviewed on OpenReview:** *https://openreview.net/forum?id=ScrEUZLxPr*

## Abstract

While a key component to the success of deep learning is the availability of massive amounts of training data, medical image datasets are often limited in diversity and size. Transfer learning has the potential to bridge the gap between related yet different domains. For medical applications, however, it remains unclear whether it is more beneficial to pre-train on natural or medical images. We aim to shed light on this problem by comparing initialization on `ImageNet` and `RadImageNet` on seven medical classification tasks. Our work includes a replication study, which yields results contrary to previously published findings. In our experiments, ResNet50 models pre-trained on `ImageNet` tend to outperform those trained on `RadImageNet`. To gain further insights, we investigate the learned representations using Canonical Correlation Analysis (CCA) and compare the predictions of the different models. Our results indicate that, contrary to intuition, `ImageNet` and `RadImageNet` may converge to distinct intermediate representations, which appear to diverge further during fine-tuning. Despite these distinct representations, the predictions of the models remain similar. Our findings show that the similarity between networks before and after fine-tuning does not correlate with performance gains, suggesting that the advantages of transfer learning might not solely originate from the reuse of features in the early layers of a convolutional neural network.

## 1 Introduction

Transfer learning has become an increasingly popular approach in medical imaging, as it offers a solution to the challenge of training models with limited dataset sizes. The ability to leverage knowledge from pre-trained models has proven to be beneficial in various medical imaging applications (Raghu et al., 2019; Cheplygina et al., 2019; Shin et al., 2016). Despite its widespread use, the precise effects of transfer learning on medical image classification are still heavily understudied.

While pre-training on `ImageNet` has become a common practice in medical image classification, there have been growing concerns within the medical imaging community regarding its suitability for medical imaging tasks. Medical images differ from natural images in several ways, including local texture variations as an indication of pathology rather than a clear global subject present in natural images (Raghu et al., 2019). Additionally, medical datasets are smaller in size, have fewer classes, have higher resolution compared to `ImageNet`, and go beyond 2D. These differences between natural and medical image datasets have led to the argument that `ImageNet` may not be the optimal solution for pre-training in medical imaging due to the well-known performance degradation effect caused by domain shift (Kondrateva et al., 2021). This has led

to increased efforts to explore alternative solutions for pre-training, such as using existing medical datasets (El-Nouby et al., 2021), their alterations (Kataoka et al., 2020; Asano et al., 2020), and creating new medical image datasets specifically designed for pre-training, such as `RadImageNet` (Mei et al., 2022).

Recent studies have challenged the conventional wisdom that the source dataset used for pre-training must be closely related to the target task in order to achieve good performance. Evidence has emerged suggesting that the source dataset may not have a significant impact on the performance of the target task and we can pre-train on any real large-scale diverse data (Gavrikov & Keuper, 2022a;b). Further, more evidence suggests that `ImageNet` leads to the best transfer performance in terms of accuracy, as `ImageNet` not only boosts the performance (Ke et al., 2021) but also is a better source than medical image datasets (Brandt et al., 2021). This is likely due to the focus on texture in `ImageNet` models (Geirhos et al., 2022), which has been hypothesized to be an important cue for medical image classification.

While `RadImageNet` has demonstrated `ImageNet`-level accuracy (Mei et al., 2022) on radiology image classification, it remains uncertain whether it leads to improved representations when applied to medical target datasets. Furthermore, it is essential to understand the broader implications of source datasets beyond their effect on target task performance, in order to enable practitioners to make more informed decisions when selecting a source dataset.

In light of the ongoing debate on the choice of source dataset for medical pre-training, we set out to investigate this with a series of systematic experiments on the difference of representations learned from natural (`ImageNet`) and medical (`RadImageNet`) source datasets on a range of (seven) medical targets. Our main contributions are:

- We extend the work presented in Mei et al. (2022) by doing a replication study of four of their seven experiments (derived from three small medical targets: `breast`, `thyroid`, and `knee` datasets) and adding four additional medical imaging target datasets. Contrary to the findings in (Mei et al., 2022), we observe that in most cases, models pre-trained on `ImageNet` tend to perform better than those trained on `RadImageNet`. However, it is important to note that this discrepancy does not necessarily indicate the superiority of one source dataset over the other. Rather, it emphasizes the sensitivity of transfer performance to the choice of model architecture and hyperparameters.

- We investigate the learned intermediate representations of the models pre-trained on `ImageNet` and `RadImageNet` using Canonical Correlation Analysis (CCA) (Raghu et al., 2017; 2019). Our results indicate that the networks may converge to distinct intermediate representations, and these representations appear to become even more dissimilar after fine-tuning. Surprisingly, despite the dissimilarity in representations, the predictions of these networks are similar. This suggests that when using transfer learning, it is important to evaluate other desirable model qualities for medical imaging applications beyond performance, such as robustness to distribution shift or adversarial attacks.

- Our findings demonstrate that model similarity before and after fine-tuning is not correlated with the improvement in performance across all layers. This suggests that the benefits of transfer learning may not arise from the reuse of features in the early layers of a convolutional neural network.

- We make our code and experiments publically available on Github[1].

## 2 Related work

### 2.1 Pre-training on different data

Transfer performance degradation due to distribution shift is a known problem. This issue is particularly relevant in scenarios where the availability of large-scale in-domain supervised data for pre-training is limited. In light of this, a line of research has emerged that challenges the common practice of pre-training on `ImageNet` and instead explores pre-training on different in-domain source datasets:

---

[1] https://github.com/DovileDo/revisiting-transfer

**Training on target data.** El-Nouby et al. (2021) have shown the potential of using denoising autoencoders for pre-training on target data in a self-supervised manner. Although this approach has yielded promising results, the experiments were conducted using natural image target datasets. Even the smallest target dataset consisted of 8,000 images, making it unclear how well this approach would translate to the domain of medical imaging where the availability of large-scale target data is often limited.

**Synthetic data** has been shown to be a viable alternative to real-world data for pre-training, particularly in domains where labeled data is scarce. Kataoka et al. (2020) have explored pre-training on gray-scale automatically generated fractals with labels and showed that this approach can generate unlimited amounts of synthetic labeled images, although it does not surpass the performance of pre-training on `ImageNet` in all cases.

**Self-supervised learning** is a strategy employed to learn data representations. He et al. (2022) proposed to mask random patches of the input image and reconstruct the missing pixels. MAE reconstruct missing local patches but lacks the global understanding of the image. To overcome this shortcoming, Supervised MAE (SupMAE) (Liang et al., 2022) were introduced. SupMAE include an additional supervised classification branch to learn global features from golden labels. Recently, MAE has been leveraged for medical image classification and segmentation (Zhou et al., 2022).

**Data augmentation** is often used to increase dataset size. Asano et al. (2020) showed that it can be used to scale a single image for self-supervised pre-training. However, this approach still falls short of using real diverse data. Even with millions of unlabeled images, it cannot fully bridge the gap between fully-supervised and self-supervised pre-training for deeper layers of a CNN.

Although aforementioned work in general computer vision has demonstrated the potential of synthetic and augmented data, the importance of large-scale labeled source datasets remains strong, particularly `ImageNet` in medical imaging (Brandt et al., 2021; Wen et al., 2021) despite its out-of-domain nature for medical targets. Recently, Mei et al. (2022) have demonstrated that `RadImageNet`, a large-scale dataset of radiology images similar in size to `ImageNet`, outperforms `ImageNet` on radiology target datasets. To further these findings, we investigate the potential of pre-training on the `RadImageNet` on a range of modalities that were not included in Mei et al. (2022) experiments, such as X-rays, dermoscopic images, and histopathological scans.

## 2.2 Effects of transfer learning

Transfer learning is a useful method for studying representations and generalization in deep neural networks. Yosinski et al. (2014) defined the generality of features learned by a convolutional layer based on their transferability between tasks. They analyzed representations learned in `ImageNet` models and found that the early layers form general features resembling Gabor filters and color blobs, while deeper layers become more task-specific. More recently, Raghu et al. (2019) studied feature reuse in medical imaging using transfer learning and found that this reuse is limited to the lowest two convolutional layers. Besides feature reuse, they demonstrated that the scaling of pre-trained weights can result in significant improvement in convergence speed.

Instead of investigating the transferability of weights at different layers, Gavrikov & Keuper (2022b) investigated the distributions of convolution filters learned by computer vision models and found that they only exhibit minor variations across various tasks, image domains, and datasets. They noted that models based on the same architecture tend to learn similar distributions when compared to each other, but differ significantly when compared to other architectures. The authors also discovered that medical imaging models do not learn fundamentally different filter distributions compared to models for other image domains. Based on these findings, they concluded that medical imaging models can be pre-trained with diverse image data from any domain.

Our work extends previous studies on the effects of pre-training by characterizing the representations learned from `ImageNet` and `RadImageNet` and investigating the implications of the source dataset on the learned representations. Our results provide additional evidence that the source domain may not be of high importance for pre-training medical imaging models, as we observe that even though `ImageNet` and `RadImageNet` pre-trained models converge to distinct hidden representations, their predictions are still similar.

## 3 Method

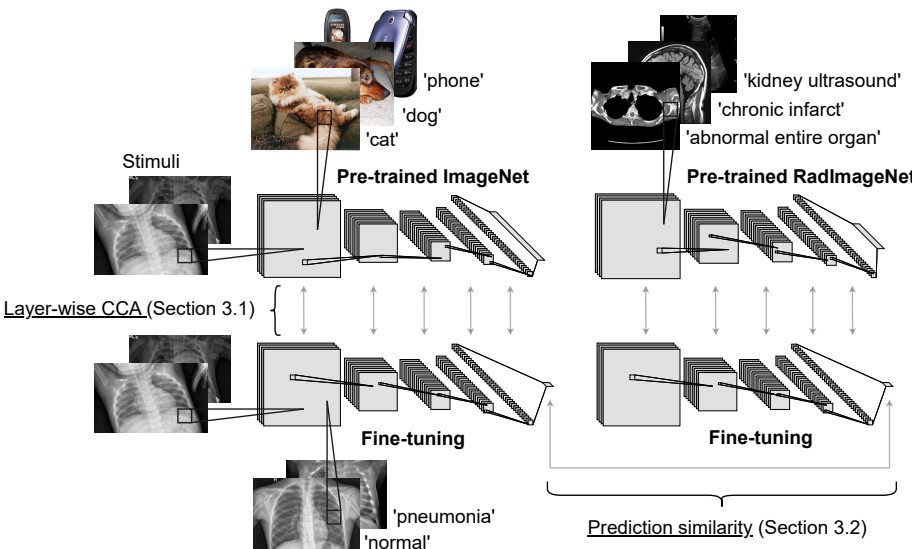

Figure 1: Overview of the experimental setup. Publicly available pre-trained `ImageNet` and `RadImageNet` weights are fine-tuned on medical targets. Model similarity is evaluated by comparing network activations over sampled stimuli images from target datasets using both CCA (described in Section 3.1) and prediction similarity (Section 3.2).

We outline our overall method in Figure 1. We fine-tune publicly available pre-trained `ImageNet` and `RadImageNet` weights on medical target datasets and quantify the model similarity by comparing the network activations over a sample of images from the target datasets using two similarity measures, Canonical Correlation Analysis (CCA, Section 3.1) and prediction similarity (Section 3.2).

### 3.1 Canonical Correlation Analysis

CCA (Hotelling, 1936) which is a statistical method used to analyze the relationship between two sets of variables.

Let $\mathbf{X}$ be a dataset $\mathbf{x}_1, \mathbf{x}_2, ..., \mathbf{x}_n$ of $n$ data points, all consisting of $p$ variables, and $\mathbf{Y}$ – a dataset of $n$ data points $\mathbf{y}_1, \mathbf{y}_2, ..., \mathbf{y}_n$ and $q$ variables. CCA seeks to find the transformation matrices $\mathbf{A}$ and $\mathbf{B}$ that linearly combine the initial variables $p$ and $q$ in the datasets $\mathbf{X}$ and $\mathbf{Y}$ into $\min(p,q)$ canonical variables $\mathbf{X}\mathbf{a}^i$ and $\mathbf{Y}\mathbf{b}^i$ such that the correlation between these canonical variables is maximized:

$$\mathbf{a}^i, \mathbf{b}^i = \operatorname*{argmax}_{\mathbf{a}^i, \mathbf{b}^i} \operatorname{corr}(\mathbf{X}\mathbf{a}^i, \mathbf{Y}\mathbf{b}^i)$$
$$\text{subject to } \forall_{j<i} \ \mathbf{X}\mathbf{a}^i \perp \mathbf{X}\mathbf{a}^j$$
$$\forall_{j<i} \ \mathbf{Y}\mathbf{b}^i \perp \mathbf{Y}\mathbf{b}^j$$

The restrictions ensure that the canonical variables are orthogonal. This can be solved by defining substitutions $\bar{\mathbf{A}} = \boldsymbol{\Sigma}_X^{1/2}\mathbf{A}$ and $\bar{\mathbf{B}} = \boldsymbol{\Sigma}_Y^{1/2}\mathbf{B}$ obtaining:

$$\bar{\mathbf{A}}, \bar{\mathbf{B}} = \underset{\bar{\mathbf{A}},\bar{\mathbf{B}}}{\arg\max} \operatorname{tr}(\bar{\mathbf{A}}^{\top} \mathbf{\Sigma}_X^{-1/2} \mathbf{\Sigma}_{XY} \mathbf{\Sigma}_Y^{-1/2} \bar{\mathbf{B}})$$

$$\text{subject to } \bar{\mathbf{A}}^{\top} \bar{\mathbf{A}} = \mathbf{I}$$

$$\bar{\mathbf{B}}^{\top} \bar{\mathbf{B}} = \mathbf{I}$$

Because of the orthogonality constraints, the solution is found by decomposing $\mathbf{\Sigma}_X^{-1/2} \mathbf{\Sigma}_{XY} \mathbf{\Sigma}_Y^{-1/2}$ into left and right singular vectors using singular value decomposition.

**Layer-wise model similarity**. Raghu et al. (2017) proposed the use of CCA for comparing representations learned by neural networks. CCA's invariance to linear combinations makes it suitable for comparing the representations learned by different models as the layer weights in neural networks are combined before being passed on (Morcos et al., 2018).

Raghu et al. (2019) used CCA to examine representations in medical imaging models. In order to maintain consistency with Raghu et al. (2019) results, we adopt the same approach of applying CCA to CNNs and use an open source CCA implementation by Raghu et al. (2017) available on GitHub[2].

In our case, $\mathbf{X}$ and $\mathbf{Y}$ are same-level layer activation vectors over $n$ stimuli images sampled from a target dataset, in two models with different initializations. We extract these intermediate representations and use them as input to CCA to project the representations onto a common space, where the correlation between the projections is maximized. This common space can be thought of as a shared representation that captures the common patterns of activity across the compared networks. Then, layer-wise similarity at layer $L$ is the average of the correlations between the canonical variables:

$$\rho_L = \sum_{i=1}^{p} \operatorname{corr}(\mathbf{X}\mathbf{a}^i, \mathbf{Y}\mathbf{b}^i) \tag{1}$$

Intermediate representations extracted from CNNs are of shape $(n, h_L, w_L, p_L)$, where $h_L$, $w_L$ are the layer spatial dimensions and $p_L$ is the number of channels in the layer. These representations are reshaped into $\mathbf{X}$ and $\mathbf{Y}$ matrices of shape $(n \times h_L \times w_L, p_L)$. As CCA is sensitive to the shape of the input matrices and the shapes vary across the layers within a network, we sample $n$ and $p_L$, such that $n \times h_L \times w_L \approx 20,000$ and $p_L = 64$, and then calculate layer similarity $\rho_L$. This is repeated five times and the final layer similarity is obtained by averaging layer similarities $\rho_L$.

## 3.2 Prediction similarity

We calculate prediction similarity as described in Mania et al. (2019). A model mistake is defined as $q_f(x, y) = \mathbf{1}_{\mathrm{f(x)} \neq \mathrm{y}}$, where $x$ is an image from the test set, $y$ is its label and $f$ is a network fine-tuned on the target training set. Then the prediction similarity of two networks $f_{\texttt{ImageNet}}$ and $f_{\texttt{RadImageNet}}$, fine-tuned on the same target dataset, is:

$$\mathbb{P}(q_{f_{\texttt{ImageNet}}}(x, y) = q_{f_{\texttt{RadImageNet}}}(x, y)) \tag{2}$$

Therefore, the prediction similarity is the probability that two networks will make the same errors. To gauge the prediction similarities between `ImageNet` and `RadImageNet` models, we compare them to the prediction similarity of two classifiers with the same accuracy as `ImageNet` and `RadImageNet` models but otherwise random predictions. If the mistakes made by two models with accuracy $a_1$ and $a_2$ are independent, the similarity of their predictions is equal to $a_1 a_2 + (1 - a_1)(1 - a_2)$.

---

[2]`https://github.com/google/svcca`

# 4 Experimental setup

## 4.1 Datasets

**Source.** We use publicly available pre-trained `ImageNet` (Deng et al., 2009) and `RadImageNet` (Mei et al., 2022) weights as source tasks in our experiments.

**Target.** We investigate transferability to several medical target datasets. In particular, to five radiology `RadImageNet` in-domain datasets, and two out-of-domain datasets in the fields of dermatology and microscopy. A representative image from each dataset can be seen in Figure 2.

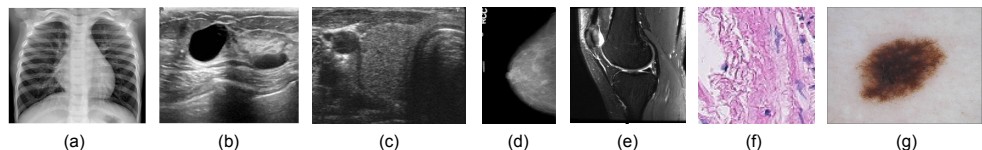

Figure 2: Example images of (a) `chest`, (b) `breast`, (c) `thyroid`, (d) `mammograms`, (e) `knee`, (f) `pcam-small`, and (g) `ISIC` datasets.

1) `Chest.` Chest X-rays (Kermany et al., 2018) dataset contains chest X-ray images from pediatric patients aged one - five years old, labeled by expert physicians with binary labels of 'normal' or 'pneumonia'. The dataset has 5,856 images, with 1,583 labeled as 'normal' and 4,273 labeled as 'pneumonia'. The image size varies, with dimensions ranging from $72 \times 72$ to $2,916 \times 2,583$ pixels.

2) `Breast`. Breast ultrasound (Al-Dhabyani et al., 2020) dataset is collected for the detection of breast cancer. The images have a range of sizes, from $190 \times 335$ to $1,048 \times 578$ pixels. The dataset is divided into three classes: normal, benign, and malignant images. However, following Mei et al. (2022), we use a binary classification of 'benign' and 'malignant' for our analysis.

3) `Thyroid.` The Digital Database of Thyroid Ultrasound Images (DDTI) (Pedraza et al., 2015) contains 480 images of size $569 \times 360$ pixels, extracted from thyroid ultrasound videos. The images have been annotated by radiologists into five categories. Following Mei et al. (2022)'s study, these categories were transformed into binary labels: 'normal' for categories (1) normal thyroid, (2) benign and (3) no suspicious ultrasound (US) feature, and 'malignant' for categories (4a) one suspicious US feature, (4b) two suspicious US features, (4c) three or four suspicious US features and (5) five suspicious features.

4) `Mammograms.` Curated Breast Imaging Subset of Digital Database for Screening Mammography (CBIS-DDSM) (Sawyer-Lee et al., 2016; Lee et al., 2017; Clark et al., 2013) is a dataset that targets breast cancer detection. It contains scanned film mammograms with pathologically confirmed labels: 'benign' (2,111 images) or 'malignant' (1,457 images) with image sizes ranging from $1,846 \times 4,006$ to $5,431 \times 6,871$ pixels.

5) `Knee.` MRNet (Bien et al., 2018) is a collection of 3D knee MRI scans. The labels for the dataset were obtained through manual extraction from clinical reports. Following Mei et al. (2022)'s study, we use extracted 2D sagittal views (1 to 3 samples per scan) amounting to a total of 4,235 'normal', 569 'ACL' (anterior cruciate ligament), and 418 'meniscal tear' images, all of size $256 \times 256$ pixels.

6) `PCam-small.` PatchCamelyon (Veeling et al., 2018) is a metastatic tissue classification dataset consisting of 237,680 colored patches extracted from histopathological scans of lymph node sections. The images are labeled as 'positive' or 'negative' based on the presence of metastatic tissue. To simulate a realistic target dataset size, a random subset of 10,000 images was created, with 5,026 positive and 4,974 negative samples.

7) `ISIC.` ISIC 2018 Challenge - Task 3: Lesion Diagnosis (Codella et al., 2019; Tschandl et al., 2018) - a dermoscopic lesion image dataset released for the task of skin lesion classification. The dataset comprises images of $600 \times 450$ pixels, which are split into seven disease categories. The dataset is unbalanced, with the class 'melanocytic nevus' having the most samples at 6,705, and the class 'dermatofibroma' having the least number of samples at 115.

Table 1: Target datasets with number of images, number of classes, image size and batch size used to fine-tune the pre-trained `ImageNet` and `RadImageNet` weights.

| Dataset | Size | Classes | Image size | Batch size |
|---|---|---|---|---|
| Chest | 5,856 | 2 | $112 \times 112$ | 128 |
| Breast | 780 | 2 | $256 \times 256$ | 16 |
| Thyroid | 480 | 2 | $256 \times 256$ | 16 |
| Mammograms | 3,568 | 2 | $224 \times 224$ | 32 |
| Knee | 5,222 | 3 | $112 \times 112$ | 128 |
| PCam-small | 10,000 | 2 | $96 \times 96$ | 128 |
| ISIC | 10,015 | 7 | $112 \times 112$ | 128 |

Due to memory constraints, we reduced the original image sizes for most of the target datasets using interpolation without image cropping. Table 1 provides details of the image sizes and number of images used for fine-tuning on each target dataset. As we used publicly available pre-trained weights images were preprocessed to align with the pre-trained weights. As per the approach in Mei et al. (2022), we normalized the images with respect to the `ImageNet` dataset. To increase the diversity and variability of the training data images were augmented during fine-tuning with the following parameters: rotation range of 10 degrees, width shift range of 0.1, height shift range of 0.1, shear range of 0.1, zoom range of 0.1, fill mode set to "nearest", and horizontal flip set to false if the target is `chest`, otherwise set to true.

### 4.2 Fine-tuning

We select ResNet50 (He et al., 2016) as the standard model architecture for our experiments. This architecture is widely adopted in the field of medical imaging and has been demonstrated to be a strong performer in various image classification tasks. We fine-tuned pre-trained networks using an average pooling layer and a dropout layer with a probability of 0.5. The hyperparameters were not tuned on any of the target datasets. Since we are targeting several tasks, we decided to fix the initial learning rate to a small value (1e-5) for all experiments, and used the Adam optimizer to adapt to each dataset. The models were trained for a maximum of 200 epochs, with early stopping after 30 epochs of no decrease in validation loss, saving the models that achieved the lowest validation loss. This was done to prevent overfitting and ensure that the models generalize well to unseen data.

In addition to full fine-tuning, we used a freezing strategy where we froze all the pre-trained weights to train the classification layer first and then fine-tuned the whole network with the same hyperparameters as above.

Models were implemented using Keras (Chollet et al., 2015) library and fine-tuned on 3 NVIDIA GeForce RTX 2070 GPU cards.

### 4.3 Evaluation

We fine-tune the pre-trained networks on each target dataset using five-fold cross-validation approach. The datasets was split into training (80%), validation (5%), and test (15%) sets. To ensure patient-independent validation where patient information is available (`chest`, `thyroid`, `mammograms`, `knee`), the target data is split such that the same patient is only present in either the training, validation or test split. We evaluate fine-tuned network performance on test set using AUC (area under the receiver operating characteristic curve). Model similarity is evaluated using CCA and prediction similarity as described in Section 3.

## 5   Results

We carried out a series of experiments to evaluate the effect of pre-training on `ImageNet` and `RadImageNet` on model accuracy and learned representations after fine-tuning on medical targets. In the following section,

Table 2: Mean AUC $\pm$ std (both $\times 100$) after fine-tuning on target datasets. Underlined is the highest mean AUC per dataset.

| | ImageNet | | RadImageNet | | Random init |
| --- | --- | --- | --- | --- | --- |
| **Target dataset** | **No Freeze** | **Freeze** | **No Freeze** | **Freeze** | **No Freeze** |
| `Thyroid` | $64.9 \pm 7.2$ | $\underline{67.8 \pm 6.2}$ | $62.7 \pm 9.1$ | $63.7 \pm 5.1$ | $64.3 \pm 7.8$ |
| `Breast` | $94.3 \pm 1.7$ | $\underline{95.1 \pm 3.6}$ | $91.0 \pm 5.2$ | $89.4 \pm 3.8$ | $85.2 \pm 1.4$ |
| `Chest` | $98.7 \pm 0.5$ | $\underline{99.0 \pm 0.3}$ | $98.7 \pm 0.3$ | $98.2 \pm 0.3$ | $97.9 \pm 0.6$ |
| `Mammograms` | $75.4 \pm 3.1$ | $\underline{77.3 \pm 1.0}$ | $74.3 \pm 2.0$ | $70.4 \pm 5.0$ | $68.3 \pm 4.4$ |
| `Knee` | $96.5 \pm 0.7$ | $97.1 \pm 1.3$ | $\underline{97.3 \pm 0.7}$ | $95.4 \pm 0.7$ | $93.2 \pm 1.6$ |
| `ISIC` | $97.4 \pm 0.3$ | $\underline{97.6 \pm 0.3}$ | $96.2 \pm 0.4$ | $95.8 \pm 0.3$ | $95.8 \pm 0.3$ |
| `Pcam-small` | $92.9 \pm 1.5$ | $\underline{94.4 \pm 0.6}$ | $87.5 \pm 1.5$ | $89.7 \pm 0.8$ | $83.2 \pm 1.1$ |

we present the results of our experiments and provide a thorough analysis of the findings. The results offer new insights into the effects of transfer learning and provide a foundation for future research in this field.

## 5.1 `ImageNet` tends to outperform `RadImageNet`

We show the AUC performances in Table 2. Overall, `ImageNet` fine-tuned after freezing the classification layer leads to the highest AUCs in six out of the seven datasets. Only `knee` reaches the highest performance with pre-traind `RadImageNet` weights, though we note that both `ImageNet` and `RadImageNet` performances were comparable for this dataset, as well as for the `chest` dataset.

Compared to Mei et al. (2022), we obtained similar AUC values for the `knee` and `breast` datasets. However, we observed a significantly lower AUC for the `thyroid` dataset. We note that Mei et al. (2022) used a subset of 349 images from the `thyroid` dataset, compared to 480 images available. Furthermore, they treated the classification of ACL and meniscal tear as separate tasks for the `knee` dataset.

We decided to include all images in the `thyroid` dataset for our experiments. Nonetheless, we were able to replicate the results reported by Mei et al. (2022) and achieved improved performance with our chosen hyperparameters for ResNet50. Specifically, we trained the models for 200 epochs with a learning rate of 1e-5, whereas Mei et al. (2022) trained their model for 30 epochs with a learning rate of 1e-4 (Appendix A). This highlights the sensitivity of transfer performance to the choice of model architecture and hyperparameters.

## 5.2 Layer-wise representations become more different after fine-tuning

In this experimental setting, we compare the similarity between `ImageNet` and `RadImageNet` against several baselines. Our baselines include the similarity of two randomly initialized networks and the similarity of fine-tuned models to randomly initialized networks.

In Figure 3 we show layer-wise `ImageNet` and `RadImageNet` CCA similarity to themselves after fine-tuning, $\text{ImageNet}^{\text{FT}}$ and $\text{RadImageNet}^{\text{FT}}$, respectively (Figure 3a), as well as layer-wise `ImageNet` and `RadImageNet` CCA similarity before and after fine-tuning (Figure 3b). `ImageNet` weights change less during fine-tuning, see Figure 3a (orange line). The two networks converge to distinct solutions after fine-tuning (both with freezing, red line on the right, and no freezing, green line), even more distinct than before fine-tuning, and their similarity is significantly lower when compared to the similarity of two random initialization. The similarity between fine-tuned networks becomes comparable to the similarity between fine-tuned networks and randomly initialized networks, particularly in the higher layers. Here we only provide results on `knee`, however we observed similar patterns for the other target datasets (Appendix B).

For experiments where pre-trained weights were initially frozen and fine-tuned after training the classification layer, we observed essentially similar trends of weight similarity as shown in Figure 3, but with higher variability between the folds of the cross-validation.

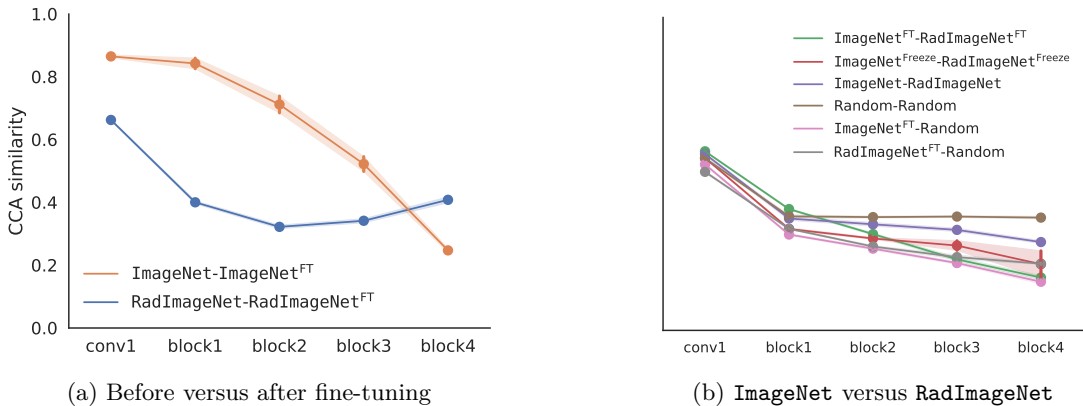

(a) Before versus after fine-tuning

(b) ImageNet versus RadImageNet

Figure 3: Layer-wise CCA similarity (Equation 1) of (a) a network to itself before and after fine-tuning on knee and (b) ImageNet to RadImageNet. ImageNet-ImageNet$^{\mathrm{FT}}$ similarity (orange line) is higher (ImageNet weights change less during fine-tuning) than RadImageNet-RadImageNet$^{\mathrm{FT}}$ similarity (blue line). ImageNet and RadImageNet are highly dissimilar after fine-tuning on the same dataset both "No Freeze" (green line), and "Freeze" (red line), even more dissimilar than before fine-tuning (purple line). Similarity of two randomly initialized networks (brown line) and ImageNet (pink line) as well as RadImageNet (gray line) similarity to randomly initialized networks are provided as baselines. Error bars present mean $\pm$ std over five-fold cross-validation.

The results of the layer-wise CCA similarity analysis reveal that ImageNet and RadImageNet converge to distinct solutions after fine-tuning on the same target dataset, to the extent that they become even more dissimilar than before fine-tuning. This outcome contradicts our expectation that the representations of the two networks would become more similar after training on the same target dataset. This discrepancy may be due to memorization of the target dataset by one or both of the networks, as suggested by the findings of Morcos et al. (2018). They found that networks trained to classify randomized labels, hence memorizing the data, tend to converge to more distinct solutions compared to networks that generalize to unseen data.

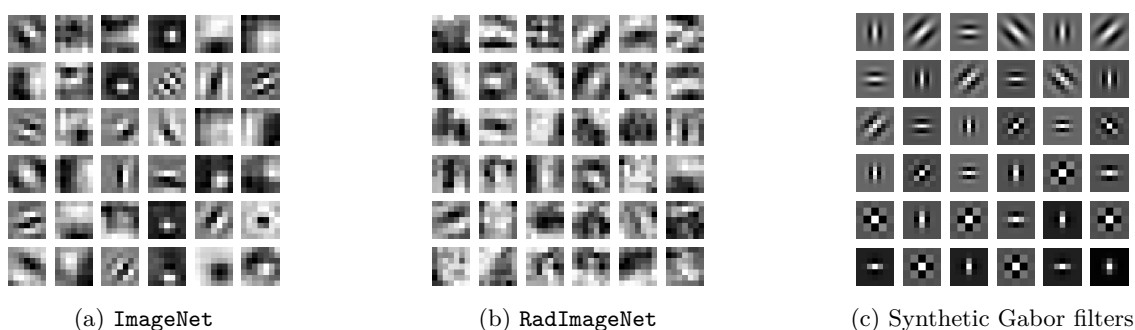

(a) ImageNet

(b) RadImageNet

(c) Synthetic Gabor filters

Figure 4: First 36 conv1 filters of ResNet50 pre-trained on (a) ImageNet and (b) RadImageNet. Observe that the filters in ImageNet have a more pronounced resemblance to (c) Gabor filters.

The stability of the representations in early layers during fine-tuning is often attributed to their capture of general features, such as edge detectors (Raghu et al., 2019; Yosinski et al., 2014), which are necessary regardless of the target domain and task. Magill et al. (2018) argued that representation similarity and generality of a layer are related, suggesting that "if a certain representation leads to good performance across a variety of tasks, then well-trained networks learning any of those tasks will discover similar representations". Contrary to this hypothesis, our findings indicate that the early layers of both ImageNet and RadImageNet exhibit similarity comparable to two randomly initialized layers. We would expect that after fine-tuning on the same task, the layers of ImageNet and RadImageNet would display greater similarity to each other than

to randomly initialized layers. Further research into general features could shed light to the learning process in early layers.

When we examine the first convolutional layer filters pre-trained on `ImageNet` and `RadImageNet` (Figure 4), we observe that the filters in `ImageNet` more closely resemble Gabor filters, while those in `RadImageNet` are more fuzzy. This is expected, as natural images often contain regular structures, such as 90 degree angles and edges, that are typically less prominent in some of radiology images, resulting in less distinct edges in the filters learned from radiology images. Interestingly, these different first layer features in both `ImageNet` and `RadImageNet`, without changing significantly during fine-tuning (Figure 3), lead to comparable performance in most cases (Table 2).

### 5.3 Networks make similar mistakes after fine-tuning

In order to further understand the similarity between `ImageNet` and `RadImageNet` pre-trained networks, we compared their predictions before and after fine-tuning on medical targets, as shown in Figure 5. Despite the networks converging to different hidden representations after fine-tuning, as evidenced in Figure 3, their predictions were found to be more similar than expected for independent predictions. The predictions before fine-tuning are less similar than after fine-tuning. We found similar behavior for both "Freeze" and "No Freeze" networks. Dataset characteristics may affect the similarity of predictions, as datasets with more than two classes (such as `knee` and `ISIC`) exhibit higher similarity in predictions before fine-tuning than independent predictions. The high variance in prediction similarity observed in the `thyroid` dataset before fine-tuning could potentially be attributed to the limited size of the dataset. It is plausible that the fine-tuning of both `ImageNet` and `RadImageNet` on the same target dataset contributes to the observed prediction similarity. However, it also suggests the possibility that the networks are learning similar misleading cues in the data, resulting in similar misclassifications.

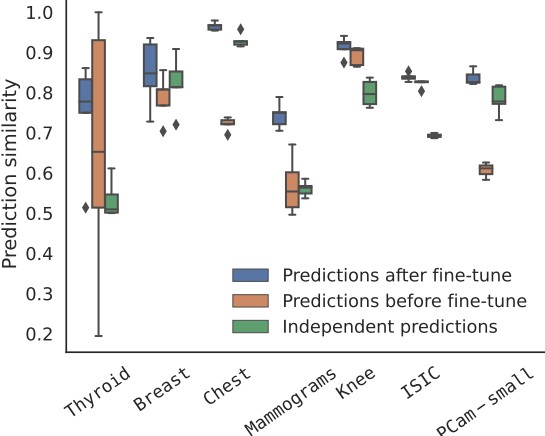

Figure 5: Prediction similarity (Equation 2) between $\text{ImageNet}^{\text{FT}}$ and $\text{RadImageNet}^{\text{FT}}$ (blue box plot), compared to prediction similarity of `ImageNet` and `RadImageNet` (orange box plot) and of two networks that would make independent mistakes (green box plot). $\text{ImageNet}^{\text{FT}}$ and $\text{RadImageNet}^{\text{FT}}$ predictions are more correlated than expected for independent predictions on average across all target datasets, with notable variation observed for predictions before fine-tuning on `thyroid` dataset.

### 5.4 Higher weight similarity associated with less AUC improvement

Our findings suggest that the benefits of transfer learning in deep neural networks may not solely stem from feature reuse, defined as layer-wise representational similarity before and after fine-tuning in the early layers (Raghu et al., 2019). Figure 6 shows that the improvement in AUC resulting from pre-training does not correlate with the layer-wise CCA similarity between the pre-trained and fine-tuned networks. Thus, models

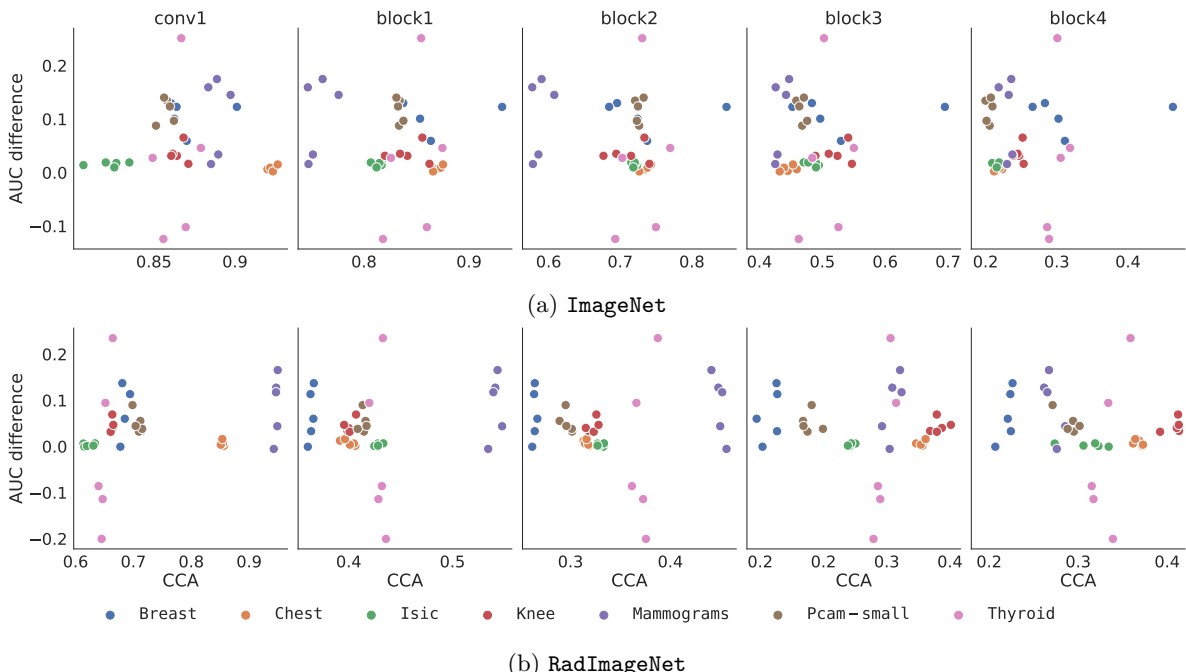

Figure 6: AUC pre-training gains over random initialization for seven target datasets vs CCA similarity before and after fine-tuning on those targets, for (a) `ImageNet` and (b) `RadImageNet`. Higher CCA similarity after fine-tuning is not associated with higher AUC gains, observed across all layers. Note that the scaling on the x-axes are different in each plot for visibility, and for `RadImageNet` the CCA similarity is lower overall.

that relied on reusing pre-trained features without adapting the representations during fine-tuning did not get higher gains in performance compared to models that underwent representation adaptation. This trend persisted across all layers for both models trained with freezing and no freezing.

Our findings align with recent results in the Natural Language Processing field which demonstrate that the benefits of pre-training are not related to knowledge transfer (Krishna et al., 2021). Additionally, our results complement the findings of Raghu et al. (2019) who showed that there are feature-independent benefits of using pre-trained weights, such as better scaling compared to random initialization. These results highlight the complex nature of transfer learning and the need for further investigation into the underlying mechanisms that drive its performance benefits.

# 6 Discussion

## 6.1 Results

In our experiments, we found that `ImageNet` initialization generally outperformed `RadImageNet` on the medical target datasets, in contrast to the earlier results reported in Mei et al. (2022). However, this highlights the sensitivity of source dataset transfer performance to the model architecture and hyperparameters, rather than the inherent superiority of one source dataset over the other. We investigated the effect of hyperparameters such as learning rate, early stopping and training epochs, and found differences in model performance. While our study did not comprehensively analyze this sensitivity, interested readers can refer to related studies, such as Raghu et al. (2019), Wen et al. (2021), and Cherti & Jitsev (2022) which offer valuable insights into the impact of model architecture and hyperparameters on transfer performance.

Contrary to our intuition about transfer learning, our analysis with CCA found that the models converged to distinct intermediate representations and that these representations are even more dissimilar after fine-tuning on the same target dataset. Despite distinct intermediate representations, model predictions on an

instance level show a significant degree of similarity. This extends Mania et al. (2019) findings which showed that `ImageNet` models are similar in their predictions even with different architectures.

## 6.2 Limitations and future work

We investigated transfer learning for two large source datasets and seven medical target datasets, and ResNet50, an architecture that is widely used for medical image analysis. Extending our experiments to further datasets and architectures, as well as other similarity measures, such as centered kernel alignment (Kornblith et al., 2019), would be valuable to further test the generalizability of our findings.

With regard to source datasets, we used `ImageNet` and `RadImageNet` because `RadImageNet`'s comparable size to `ImageNet` allowed for a unique opportunity to compare natural and medical source datasets. However, the two datasets have several differences beyond their domains. For instance, `RadImageNet` has differences in color, number of classes, and diversity in data due to its limited number of patients. To further explore the impact of these differences in greater detail, future research could consider including Ecoset (Mehrer et al., 2021), a natural image dataset with 565 basic-level categories selected to better reflect the human perceptual and cognitive experience.

Another limitation of our study is the use of a single classification metric (AUC) for evaluating performance. AUC is a commonly used metric for classification tasks in medical imaging, and useful to compare to related work. However, there might be nuances across applications where it could be important to consider alternative metrics, such as calibration (Reinke et al., 2021).

## 6.3 Recommendations

In our experiments, we only used 2D images, but these tasks are not fully representative of the medical imaging field as a whole. Researchers have hypothesized that for 3D target tasks such as CT or MRI, 3D pre-training might be a better alternative to 2D pre-training (Wong et al., 2018; Brandt et al., 2021), and studies have shown that incorporating information from the third dimension might be beneficial for performance (Yang et al., 2021; Taleb et al., 2020). However, `RadImageNet` only has 2D images, even though some of the original images are 3D. For 3D target tasks, consider comparing both 2D pre-training (e.g. via a 2.5D approach) and 3D pre-training. Pre-trained weights for 3D models are less common, but previous research has successfully used pre-training on for example YouTube videos (Malik & Bzdok, 2022).

Regarding fine-tuning with `RadImageNet`, we recommend using higher learning rates when fine-tuning compared to `ImageNet`. Furthermore, we suggest allocating more epochs in order to achieve optimal model performance.

Our results suggest that the implications of using `ImageNet` in medical image classification go beyond performance alone. In particular, there might be an issue with the memorization of spurious patterns in the data. This can potentially have consequences with respect to algorithmic bias and fairness. For example, see Gichoya et al. (2022) where `ImageNet` pre-trained networks memorize patient race. Memorization also makes a network more vulnerable to adversarial attacks (Bortsova et al., 2021). In the event that for a target application, `ImageNet` and `RadImageNet` weights are expected to lead to similar performances, it might be an advantage to select the weights which are less associated with these negative properties.

## 7 Conclusions

Transfer learning is a key strategy to leverage knowledge from the models pre-trained on large-scale datasets to deal with the challenge of small medical datasets. In this study, we investigated the transferability of two different domain sources (natural: `ImageNet` and medical: `RadImageNet`) to seven target medical image classification tasks with limited dataset size. Our results show that pre-training ResNet50 on `ImageNet` outperformed `RadImageNet` in most cases. Furthermore, we delved deeper into the learned representations after fine-tuning by using CCA and comparing the similarity of predictions. Although the models appear to converge to distinct representations, we found they made similar predictions. Lastly, we observed that higher model similarity before and after fine-tuning did not result in higher performance gains.

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

# A    Replication of results from Mei et al. (2022)

Table 3: Mean AUC $\pm$ std (both $\times 100$) after fine-tuning on two different versions of the `Thyroid` dataset: subset used in Mei et al. (2022), and full. For the results in the first row, we trained a ResNet50 with the parameters in Mei et al. (2022), specifically learning rate 1e-4 and 30 epochs. The results in the second row correspond to a ResNet50 trained with our hyperparameters specified in Section 4.2. In the third row, we show the results averaged over all models reported in the paper by Mei et al. (2022).

|  | `Thyroid` subset | | `Thyroid` full | |
| --- | --- | --- | --- | --- |
| **Experiment** | **ImageNet** | **RadImageNet** | **ImageNet** | **RadImageNet** |
| ResNet50 (Mei et al. (2022)) | $81.7 \pm 5.5$ | $85.4 \pm 4.7$ | $62.8 \pm 6.9$ | $64.3 \pm 7.7$ |
| ResNet50 (Our parameters) | $87.6 \pm 4.1$ | $85.9 \pm 3.6$ | $64.9 \pm 7.2$ | $62.7 \pm 9.1$ |
| Average over all models (Mei et al. (2022)) | $76 \pm 14$ | $85 \pm 9$ | | |

# B    Layer-wise CCA similarity

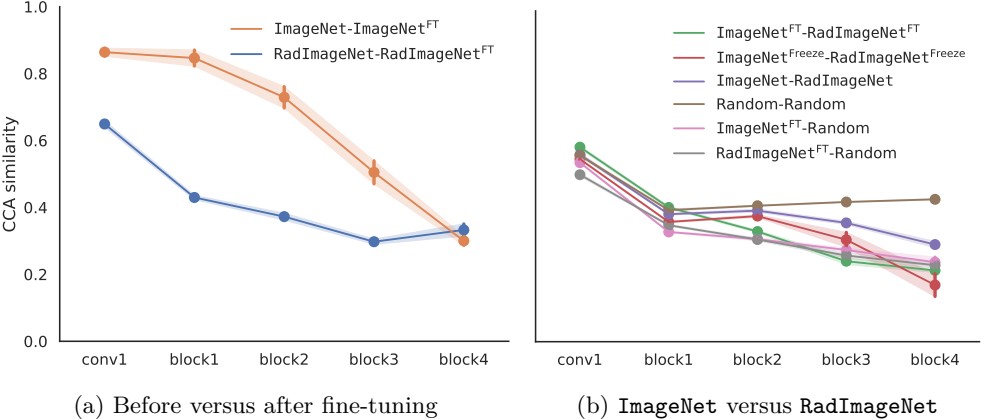

(a) Before versus after fine-tuning        (b) `ImageNet` versus `RadImageNet`

Figure 7: Layer-wise CCA similarity of networks fine-tuned on `thyroid`.

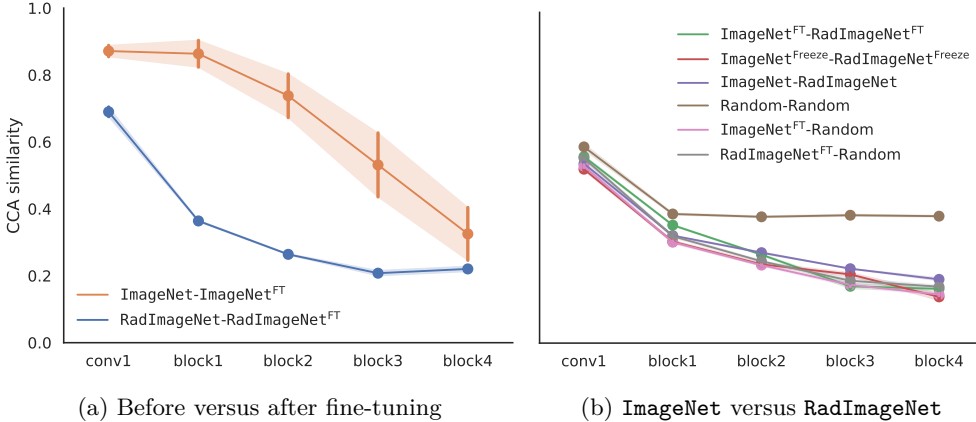

(a) Before versus after fine-tuning        (b) `ImageNet` versus `RadImageNet`

Figure 8: Layer-wise CCA similarity of networks fine-tuned on `breast`.

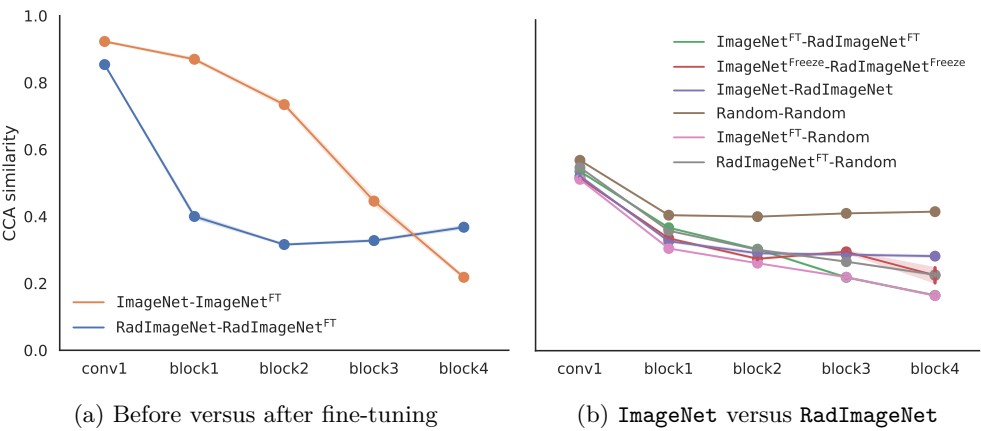

(a) Before versus after fine-tuning      (b) ImageNet versus RadImageNet

Figure 9: Layer-wise CCA similarity of networks fine-tuned on chest.

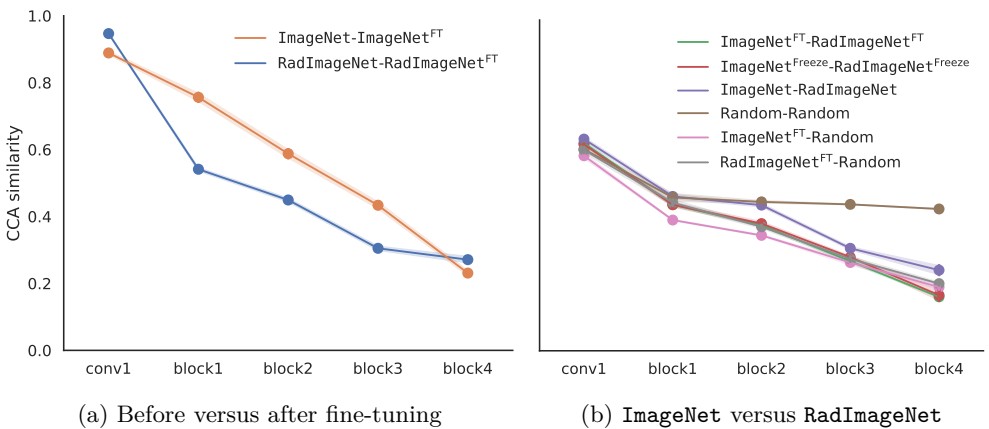

(a) Before versus after fine-tuning      (b) ImageNet versus RadImageNet

Figure 10: Layer-wise CCA similarity of networks fine-tuned on mammograms.

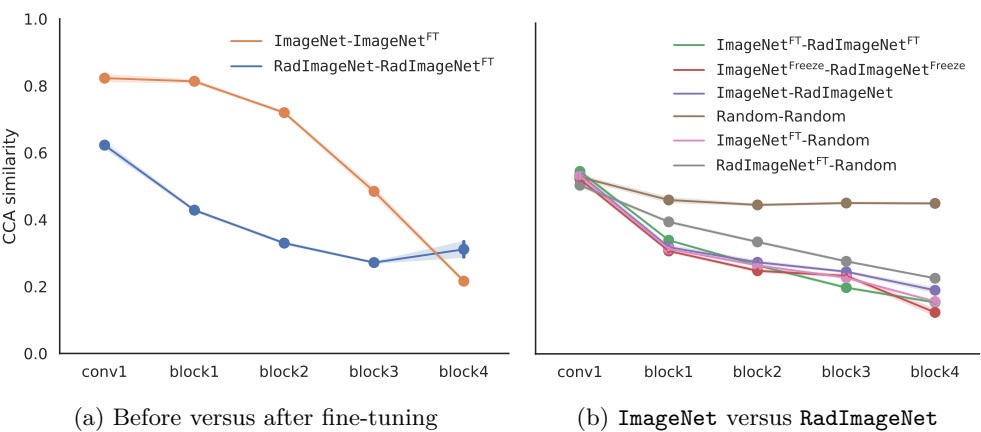

(a) Before versus after fine-tuning      (b) ImageNet versus RadImageNet

Figure 11: Layer-wise CCA similarity of networks fine-tuned on isic.

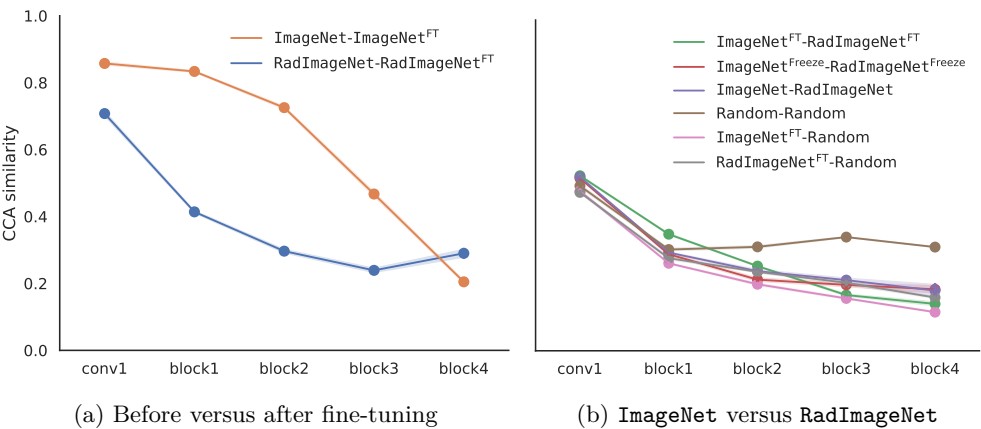

(a) Before versus after fine-tuning

(b) ImageNet versus RadImageNet

Figure 12: Layer-wise CCA similarity of networks fine-tuned on pcam-small.

