# OpenReview forum: "Revisiting Hidden Representations in Transfer Learning for Medical Imaging"
_TMLR — Accepted by TMLR_

### Review · Reviewer_3DKR · 2023-06-13

**Summary Of Contributions:**

This work presents an analysis of transfer learning in the context of medical imaging.
It aims to provide insights on the effect of pre-training datasets on the features.
This is done by studying the similarities in predictions and activations of networks that are pre-trained on imagenet and a medical imagenet variant.

**Audience:**

Yes

**Broader Impact Concerns:**

No concerns

**Claims And Evidence:**

No

**Requested Changes:**

 - Add discussion on self-supervised pre-training.
 - Explain how hyper-parameters were chosen and what dataset(s) were used to choose hyper-parameters.
 - Add details to the experiments:
    1. Report the steps that were taken to reproduce the results from (Mei et al., 2022).
    The goal should be to reproduce the aggregated results, but it should be easy to extract the performance of the individual models once results have been reproduced.
    (also, see https://github.com/BMEII-AI/RadImageNet)
    2. Provide baseline similarities to assess effects due to initialisation for pre-training or normalisation of the data.
       Also, the frozen models might be interesting for establishing baselines.
    3. Revisit the interpretation/explanation of the inconsistency results.
    4. Include prediction similarity for networks before fine-tuning (e.g. the frozen networks).
    5. The statement on feature reuse needs to be significantly weakened.
       Alternatively, more experiments and better argumentation are necesessary to justify the statement.

**Strengths And Weaknesses:**

### Strengths

 - Overall, the manuscript is well written and easy to follow.
 - This work appears to be nicely embedded into existing work.
   However, it might be that I am unaware of some of the literature.
 - The focus of this paper lies on the analysis of transfer learning.
   There are no methodological novelties and all of the used techniques are readily available.

### Weaknesses

 - There seems to be little context on self-supervised pre-training in related work.
   To the best of my knowledge, the vision community prefers self-supervised pre-training.
   Especially masked auto-encoders seem to be superior to supervised pre-training.
   Section&nbsp;2.1, §4 (data augmentation) seems to imply the opposite.
 - There is no information on how hyper-parameters were chosen.
 - The results of the experiments are rather inconclusive:
    1. _"ImageNet tends to outperform RadImageNet"_ appears to be in direct contradition to the results in Mei et al. (2022).
       This is a strong claim, but the text (section&nbsp;5.1, §2) suggests that no proper attempt was made to reproduce the results from Mei et al. (2022).
       Without a proper reproduction, the worse results might as well be due to a bug.
    2. _"representations become more different after fine-tuning"_ is hard to interpret without a proper reference.
       I believe the (dis)similarity of representations for two networks pre-trained with different seeds would be interesting.
       After all, there are multiple reasons why representations could be different, apart from the dataset on which the models were pretrained.
    3. I do not understant how the observations are _"inconsistent with the intuition that early layers capture general features"_.
       I would argue that is possible to have two different sets of features that are both general, but dissimilar.
    4. _"The fact that both [...] were fine-tuned on the same target dataset may explain this observation"_ comes without evidence.
       What do the prediction similarities before fine-tuning look like?
    5. The following statement appears to be overly stong: _"Our results challenge the [...] belief that the benefits [...] come from feature reuse."_
       If we define feature reuse as the reuse of convolutional filters, there are probably a lot of unused filters that can be removed or changed arbitrarily (e.g. Frankle et al., 2019).
       Therefore, the results merely illustrate that some features are not reused.
       However, this is just a weak form of the lottery ticket hypothesis (Frankle et al., 2019).
       Moreover, it is unclear whether the decrease in similarity effectively corresponds to not reusing features.
       E.g. if the difference in similarity is caused by changes in bias parameters or just the scale of the values,
       I would argue that the features are still being reused.

### Minor Comments

 - The statement _"performance degradation effect caused by domain shift"_ (intro, §2) is somewhat obvious, but it would be good to include a citation for reference.
 - How does sampling $n, p_L$ help with the sensitivity to shape (in section 3.1)?
   Are they sampled such that $n \cdot p_L$ is a constant?
   If yes, how do you account for changes in the spatial dimensions ($h_L$ and $w_L$)?
 - What is meant with weight CCA (section&nbsp;5.4, §1)?
   Is this CCA on the weights instead of features?
   Should this have been weighted CCA and if yes, weighted by what?
 - The overall trends in figure&nbsp;6 are clearly not linear.
   Therefore, I would refrain from including the suggestive linear fits.

### References

 - Frankle, J., & Carbin, M. (2019). The Lottery Ticket Hypothesis: Finding Sparse, Trainable Neural Networks. ICLR 7. https://openreview.net/forum?id=rJl-b3RcF7

---

> ### Author Response · Authors · 2023-06-28
> **Author Response**
>
> We would like to thank the reviewer for providing valuable and insightful comments. Below is the summary of changes made in the updated manuscript.
>
> **Regarding self-supervised**
>
> We have added the following paragraph to the Related Work section: “Self-supervised learning is a strategy employed to learn data representations. He et al. (2022) proposed to mask random patches of the input image and reconstruct the missing pixels. MAE reconstruct missing local patches but lacks the global understanding of the image. To overcome this shortcoming, Supervised MAE (SupMAE) (Liang et al., 2022) were introduced. SupMAE include an additional supervised classification branch to learn global features from golden labels. Recently, MAE has been leveraged for medical image classification and segmentation (Zhou et al., 2022).”
>
> *He, Kaiming, et al. "Masked autoencoders are scalable vision learners." Proceedings of the IEEE/CVF Conference on Computer Vision and Pattern Recognition. 2022.*
>
> *Liang, Feng, Yangguang Li, and Diana Marculescu. "SupMAE: Supervised masked autoencoders are efficient vision learners." arXiv preprint arXiv:2205.14540 (2022).*
>
> *Zhou, Lei, et al. "Self pre-training with masked autoencoders for medical image analysis." International Symposium on Biomedical Imaging 2023.*
>
> **Regarding the specific hyperparameters**
>
> We have included the justification of the hyperparameters choice in Section 4.2 Fine-tuning. The hyperparameters were not tuned on any of the target datasets. Since we are targeting several tasks, we decided to fix the starting learning rate to a small value (1e-5) for all experiments and used Adam optimizer to adapt to each dataset. We have included the suggestion of reviewer b757 and increased early stopping to 30 epochs.
>
> **Regarding 1. reproducing the results in Mei et al. (2022)**
>
> We have reproduced the results by Mei et al. (2022) with our code and included the details in Appendix A. Lower performance on Thyroid dataset compared to performance reported by Mei. et al. (2022) is due to difference in datasets, Mei. et al. (2022) used a subset
> of 349 images from the Thyroid dataset, we use all 480 images available on Kaggle (https://www.kaggle.com/datasets/dasmehdixtr/ddti-thyroid-ultrasound-images).
>
> **Regarding 2. baseline similarities**
>
> It is indeed relevant to include baseline similarities, and we have tried to do this in the original submission in Figure 3. Perhaps we did not explain the figure in enough detail, leading to some misunderstanding of the contents. We have now rephrased section 5.2 Layer-wise representations become more different after fine-tuning. We have included fine-tuned ImageNet and RadImageNet similarity to randomly initialized weights as an additional baseline and modified this section to clarify the use of “baselines” in this experimental setting In particular, we have included “In this experimental setting, we compare the similarity between ImageNet and RadImageNet against several baselines. Our baselines include the similarity of two randomly initialized networks and the similarity of fine-tuned models to randomly initialized networks.”. Frozen models (without fine-tuning) are shown in purple line in Figure 3b, also frozen models (without fine-tuning) and fine-tuned models are shown in figure 3a. We hope that this addresses the concerns raised by the reviewer regarding the baselines. However, if there is any misunderstanding on our part, we kindly request further clarification or elaboration. Your additional insights would be greatly appreciated.

---

> > ### Author Response · Authors · 2023-06-28
> >
> > **Regarding 3. Interpretation/explanation of the inconsistency results (general features)**
> >
> >  We follow the definitions in (Magill et al. 2018, Yosinski et al. 2014) about general features. We consider that a representation is composed of general features if that representation leads to good performance across a variety of tasks (Yosinski et al. 2014). Magill et al. (2018) argued that representation similarity and generality of a layer are related, suggesting that “if a certain representation leads to good performance across a variety of tasks, then well-trained networks learning any of those tasks will discover similar representations”. This previously observed behavior might be attributed to training on similar tasks.
> >
> > We have clarified the context and semantics of our observations rephrasing the following paragraph in Section 5.2 Layer-wise representation becomes more different after fine-tuning: “Magill et al. (2018) argued that representation similarity and generality of a layer are related, suggesting that "if a certain representation leads to good performance across a variety of tasks, then well-trained networks learning any of those tasks will discover similar representations". Contrary to this hypothesis, our findings indicate that the early layers of both ImageNet and RadImageNet exhibit similarity comparable to two randomly initialized layers. We would expect that after fine-tuning on the same task, the layers of ImageNet and RadImageNet would display greater similarity to each other than to randomly initialized layers. Further research into general features could shed light on the learning process in early layers.”
> >
> > *Yosinski, Jason, et al. "How transferable are features in deep neural networks Advances in Neural Information Processing Systems 27 (2014).*
> >
> > *Magill, Martin, Faisal Qureshi, and Hendrick de Haan. "Neural networks trained to solve differential equations learn general representations." Advances in Neural Information Processing Systems 31 (2018).*
> >
> >  **Regarding 4. prediction similarity for networks before fine-tuning**
> >
> > We have included the prediction similarity of the networks before fine-tuning (freeze) and updated the plot (Figure 5) in Section 5.3 Networks make similar mistakes after fine-tuning. We see that the predictions before fine-tuning are less similar than after fine-tuning. We found similar behavior between the “freeze” and “no freeze” networks”.
> >
> > **Regarding 5. feature reuse**
> >
> > We agree that it is interesting to explore feature reuse in the context of the Lottery ticket hypothesis, however, for now, given limited time, we have updated the manuscript to bring more clarity. We use the term feature reuse as defined in Raghu et al. (2019) as layer-wise representational similarity before and after fine-tuning. We have now included changes in the abstract and Section 5.4 to adjust the context of our claim about feature reuse for medical images and explain better the insights of our analysis.
> >
> > In the abstract we have rephrased “Our findings challenge the notion that transfer learning is effective due to the reuse of general features in the early layers of a convolutional neural network” to “Our findings show that the similarity between networks before and after fine-tuning does not correlate with performance gains, suggesting that the advantages of transfer learning may not solely originate from the reuse of features in the early layers of a convolutional neural network.”.
> >
> > Moreover, we have rephrased section 5.4 to ‘’Our findings suggest that the benefits of transfer learning in deep neural networks may not solely stem from feature reuse, defined as layer-wise representational similarity before and after fine-tuning in the early layers
> > (Raghu et al., 2019). Figure 6 shows that the improvement in AUC resulting from pre-training does not correlate with the layer-wise CCA similarity between the pre-trained and fine-tuned networks. Thus, models that relied on reusing pre-trained features without adapting the representations during fine-tuning did not get higher gains in performance compared to models that underwent representation adaptation. This trend persisted across all layers for both models trained with freezing and no freezing.’’
> >
> > We have also modified the introduction and conclusions accordingly.
> >
> > *Raghu et al. “Transfusion: Understanding Transfer Learning for Medical Imaging”. NeurIPS, 2019.*

---

> > > ### Author Response · Authors · 2023-06-28
> > >
> > > **Regarding Minor Comments**
> > >
> > > * We have added a reference to the “performance degradation effect caused by domain shift" (intro, §2)
> > >
> > > * How does sampling n,p_L help with the sensitivity to shape (in section 3.1)? Are they sampled such that n⋅p_L is a constant? If yes, how do you account for changes in the spatial dimensions (h_L and w_L)? Following the implementation of CCA in Raghu et al. (2019) we sample n such that $n \cdot h_L \cdot w_L \approx 20,000$.  To keep the layers comparable within a network, we sample $p_L$ channels in all the layers, $p_L = 64$ in our case. This clarification is added to the section 3.1.
> > >
> > > * The CCA is not weighted, it is the same CCA as used throughout the paper, we have changed this in the text to layer-wise CCA in (section 5.4, §1).
> > >
> > > * We have removed the regression line from Figure 6.
> > >
> > > *Raghu et al. “Transfusion: Understanding Transfer Learning for Medical Imaging”. NeurIPS, 2019.*

---

### Review · Reviewer_b757 · 2023-06-15

**Summary Of Contributions:**

The authors argue that for transfer learning in medical diagnosis, there is no advantage in pretraining on medical images over natural images. While pretraining on medical and natural images yield dissimilar representations in terms of CCA, the models tend to make similar errors.

**Audience:**

Yes

**Claims And Evidence:**

No

**Requested Changes:**

Since the authors of RadImageNet made their dataset public and used publicly available models, I request that the authors make their reproduced results more comparable to the originally reported ones. There will be some minor discrepancies, but they should not be on the order of 0.52 vs. 0.85 AUROC.

I understand that it may be computationally infeasible to fully reproduce the results from RadImageNet, but I believe compromises can be made. E.g. for the thyroid task I highlighted above, the dataset only consists of 480 images, so fine-tuning on this dataset would not take long, especially since the authors of the original paper only fine-tuned for 30 epochs.

I would start by exactly matching every aspect of the original author's experimental setup that they reported (under computational constraints). I suspect something may have gone with image preprocessing, so I would double check that first. Also, this paper reports early stopping with a patience of three epochs. This is much too aggressive on small datasets where the loss has high variance, and will most certainly lead to underfitting. Mirroring the original authors' setup as much as you can will eliminate potential issues such as these.

**Strengths And Weaknesses:**

Strengths:

The research question is highly relevant and interesting, and the authors' message would be impactful if it were correct, since it directly contradicts those reported in RadImageNet. However, I strongly suspect the results in this paper are incorrect.

Weaknesses:

There appears to be a critical issue with implementation of the models that use the RadImageNet pretrained weights. For example, the authors report 0.52 mean AUROC on the Thyroid task when using the RadImageNet pretrained weights. The authors of RadImageNet used the same initialization to evaluate on the same Thyroid task, and report 0.85 AUROC.

The authors acknowledge that their reproduced RadImageNet-pretrained results are worse than the originals, and note some differences in the experimental setup that make a direct comparison difficult (Section 5.1). However, the disparity between 0.85 and 0.52 AUROC is too significant to attribute to these differences in experimental setup, and I strongly suspect that the results in this paper are incorrect.

---

> ### Author Response · Authors · 2023-06-20
> **Author Response**
>
> We appreciate and understand the concern raised by the reviewer.
>
> Regarding the results on Thyroid, we noticed that Mei et al. (2022) use a subset of the Thyroid dataset (349 images), while we used all 480 images publicly available on Kaggle (https://www.kaggle.com/datasets/dasmehdixtr/ddti-thyroid-ultrasound-images).  After running our code on the subset, we reproduced the results of Mei et al. We report both the full dataset, and the Mei et al. subset performances below (this table will be included in the manuscript as Table 2):
>
> | Hyperparameters  | Thyroid      | subset       | Thyroid      | full         | Reported by  | Mei et al., (2022)        |
> |------------------|--------------|--------------|--------------|--------------|--------------|-------------|
> |                  | ImageNet     | RadImageNet  | ImageNet     | RadImageNet  | ImageNet     | RadImageNet |
> | Mei et al., (2022) | 81.7 +/- 4.9 | 85.4 +/- 4.2 | 62.8 +/- 6.1 | 64.3 +/- 6,9 | 76 +/- 14    | 85 +/- 9    |
> | Our              | 87.6 +/- 3.6   | 85.9 +/- 3.2 | 64.4 +/- 6.0 | 62.7 +/- 8.2 |              |             |
>
> The first row reflects fine-tuning using hyperparameters reported by Mei et al. (2022), specifically 30 epochs and a learning rate of 0.0001. Additionally, we utilized our previously used hyperparameter combination of 200 epochs with a learning rate of 0.00001. Following the advice from the reviewer, we adjusted the patience parameter and set it to 30 instead of 3 epochs. This adjustment has significantly improved the results of both models.
>
> We agree that the previous early stopping criterion was very low, and we have increased it to avoid underfitting of our models.  We are currently in the process of running additional experiments with this adjustment to validate our findings, and will add these to the manuscript in the coming weeks.

---

> > ### Author Response · Authors · 2023-06-28
> >
> > We have now uploaded a revised version of the manuscript that includes updated fine-tuning results.

---

> > > ### Author Response · Authors · 2023-06-29
> > >
> > > In summary, the differences between our results and Mei et al. (2022) results were due to a different subset of data. We were able to replicate the Mei et al. (2022) results and found that lower learning rates were beneficial for performance. We have increased the patience parameter from 3 to 30 epochs for early stopping and retrained all models. The updated manuscript, specifically Table 2, shows improved performance across all datasets. Notably, ImageNet with freezing exhibited best performance on 6 out of 7 datasets, while RadImageNet performed best on the Knee dataset. These outcomes have not altered our overall conclusions.

---

### Review · Reviewer_LDEB · 2023-06-19

**Summary Of Contributions:**

This study examines the impact of pre-training on medical image classification using both natural (ImageNet) and medical (RadImageNet) source datasets. The authors perform experiments on seven medical tasks and analyze the learned representations using Canonical Correlation Analysis (CCA). The results indicate that the models trained on ImageNet and RadImageNet datasets develop different intermediate representations, which further diverge after fine-tuning. However, despite these differences, the models still yield similar predictions. This research challenges the notion of feature reuse in transfer learning and demonstrates a negative correlation between model similarity and performance improvements.


**Audience:**

Yes

**Claims And Evidence:**

Yes

**Requested Changes:**

The reliability of feature similarity is the foundation for many results in this article. However, this paper only utilizes CCA to analyze feature similarity and examines features from only one network architecture, which weakens the persuasiveness of the conclusions.
Some data in Table 2 exhibits significant variance (e.g., the "RadImagenet Freeze" entry for the Breast dataset), suggesting that it may not have been properly trained. It is also necessary to provide detailed explanations of the specific hyperparameters used for data augmentation and regularization terms.
Table 4 should include visualizations of the Gabor filters for comparison purposes.
The contribution section should extract more specific and guiding conclusions regarding fine-tuning in medical tasks, rather than solely refuting previous arguments. Additionally, it would be insightful for the research community if the shortcomings of fine-tuning based on RadImageNet were alleviated with corresponding improvements.

**Strengths And Weaknesses:**

Strengths:
This paper provides a comprehensive investigation of the difference in representations learned from natural (ImageNet) and medical (RadImageNet) source datasets for diverse medical image classification tasks.
The paper compares intermediate representations before and after fine-tuning, providing insights into how transfer learning affects learned representations and subsequent performance on medical image classification tasks. Additionally, the analysis on Conv1 filter visualization and its connection with the Gobar filter is interesting.
The paper is written in a clear and concise manner, making it easy to understand the research objectives and findings.

Weaknesses:
Applying CCA for analyzing representation similarity is not considered highly innovative and may have certain limitations. For instance, CCA is unable to accurately measure meaningful similarities when dealing with representations of higher dimensions than the number of data points [1]. As an alternative, it is recommended to incorporate additional feature similarity metrics, such as CKA (Centered Kernel Alignment) [1], which is widely recognized as one of the most popular metrics capable of addressing some of the limitations associated with CCA.
The study only investigated representation similarity within a single model architecture. It is proper to incorporate additional model architectures, such as the publicly available DenseNet and Inception with RadImageNet pre-trained weights, to measure representation similarity. This will enhance the persuasiveness of the conclusions drawn in the paper.
The argument challenging the feature reuse claim lacks persuasiveness. The correlation between CCA similarity and AUC gain is not apparent (Fig. 6), and this lack of clarity is even more pronounced in the early layers (Fig. 6 conv1).

[1] Kornblith, Simon, et al. "Similarity of neural network representations revisited." International Conference on Machine Learning. PMLR, 2019.

---

> ### Author Response · Authors · 2023-06-28
> **Author Response**
>
> We would like to thank the reviewer for providing valuable and insightful comments. Below is the summary of changes made in the updated manuscript.
>
> **Regarding the use of one similarity measure and one model**
>
> We understand the concern of the reviewer. However, we believe that CCA is an appropriate choice and would like to further clarify the reasons:
> * The issue of the dimensions of representations and number of data points is addressed by Raghu et al. (2019) in their implementation of CCA. According to that, we sample images n such that the number of data points (actually $h \cdot w \cdot n$ for convolutional layers) is $n \cdot h_L \cdot w_L \approx 20,000$.  To keep the layers comparable within a network, we sample $p_L$ channels in all the layers, such that $p_L = 64$ in our case. This clarification is added to the Section 3.1 Canonical Correlation Analysis.
> * CCA has been employed to analyze layer similarity in studies on medical images (Raghu et al. 2019, Wen et al. 2021). Wide use of linear CKA has been recently criticized by Davari et al. 2022.
> * We agree that different properties of other similarity measures and/or adding more networks might bring more insight, but it is not feasible for us to complete these experiments and analyze the results for all datasets within the required response time of TMLR. For now, we have emphasized in our discussion the limitations and generalizability of our results. We leave this for future work.
>
> *Raghu, Maithra, et al. "Transfusion: Understanding transfer learning for medical imaging." Advances in Neural Information Processing Systems 32 (2019).*
>
> *Wen, Yang, et al. "Rethinking pre-training on medical imaging." Journal of Visual Communication and Image Representation 78 (2021): 103145.*
>
> *Davari, MohammadReza, et al. "Reliability of CKA as a Similarity Measure in Deep Learning." International Conference on Learning Representations (2023).*
>
> **Regarding Table 2**
>
> We have retrained the models using patients of 30 epochs for early stopping instead of 3 epochs and updated the results in Table 2. We observe lower variance for all target datasets.
>
> **Regarding the specific hyperparameters**
>
> We have specified the parameters for data augmentation in section 4.1 Datasets and included the justification of the hyperparameters choice in Section 4.2 Fine-tuning. Since we are targeting several tasks, we decided to fix the starting learning rate to a small value (1e-5) for all experiments and used Adam optimizer to adapt to each dataset, saving the models that reach lowest validation loss during fine-tuning.
>
> **Regarding the Gabor filters**
>
> We have now added the visualization of the Gabor filters to Figure 4.
>
> **Regarding contributions and shortcomings**
>
> We have added structure and extended the discussion section to better distinguish the shortcomings of our study. The discussion is now divided into a discussion of the results, a dedicated limitations/future work section, and recommendations for medical imaging community based on the insights into transfer learning. We have also updated the contribution section.

---

### Decision · Action_Editors · 2023-08-29

**Recommendation:** Accept with minor revision

**Comment:**

This is a borderline case as the reviewers are leaning towards rejection given the lack of support for some of the conclusions. However, the TMLR acceptance criteria are satisfied.

**Audience:**

Some members in medical AI might find the work of interest, although I believe a venue like CHIL or ML4H might have been more appropriate.

**Claims And Evidence:**

The reviewers found the paper interesting and well written. The results show that, contrary to the work of Mei et al., pre-training with ImageNet leads to similar downstream performance than pre-training with RadImageNet. The authors use a commonly accepted similarity metric to investigate differences in feature representation between the different pre-trainings and fine-tunings.

The authors have addressed a major issue related to reproducing the results of Mei et al., and overall replied satisfactorily to most comments. However, the reviewers question the strength and generalizability of the results as these were obtained from a single architecture and single similarity metric.

I believe these are important points, especially given the recent/concurrent work of Cherti and Jitsev (1). I would encourage the authors to re-focus the work as a replication study and downplay some of their other claims given the concerns of the reviewers.

(1) Effect of Pre-Training Scale on Intra- and Inter-Domain Full and Few-Shot Transfer Learning for Natural and Medical X-Ray Chest Images. https://arxiv.org/pdf/2106.00116.pdf

**Resubmission Of Major Revision:**

The authors may consider submitting a major revision at a later time.

---

> ### Author Response · Authors · 2023-09-07
>
> We would like to take this opportunity to thank the editorial team and the reviewers for their invaluable time and insightful comments.
>
> We have uploaded both the camera-ready version and the revised paper. In this latest iteration, all alterations made in the camera-ready version since the previous revision are denoted in blue. To offer a concise overview, here is a summary of the changes:
> * We have revised the abstract to include mention of the replication study.
> * Our claims have been rephrased in the abstract, introduction, and conclusions.
> * We have cited the work of Cherti and Jitsev in the discussion.
>
> We hope this aligns with what was envisioned for the revision. Once again, we extend our gratitude for all the valuable feedback that has been instrumental in refining our work.